# Interactions of SARS-CoV-2, influenza and respiratory syncytial virus influence epidemic timing and risk
Yonghong Liu[1,6], Xiaoli Wang[1,2,3,6], Mengyao Li[1], Eimear Cleary[4], Zhifeng Cheng[4], Wenbin Zhang [4], Ying Shen[1,2], Hui Yao[1], Jiatong Han[4], Nick W. Ruktanonchai[5], Andrew J. Tatem [4], Shengjie Lai [4] ✉, Quanyi Wang [1,2,3] ✉ & Peng Yang[1,2,3] ✉

## Abstract

**Background** Interactions between SARS-CoV-2, influenza virus, and respiratory syncytial virus (RSV) at the population level remain poorly understood. This study aimed to quantify potential interactions among these viruses and assess their influence on transmission dynamics.

**Methods** We analyzed weekly surveillance data on SARS-CoV-2, influenza A and B viruses (IAV and IBV), and RSV from seven regions from October 2021 to May 2024. Distributed lag nonlinear models within a spatiotemporal Bayesian hierarchical framework were used to assess the exposure-lag-response associations among virus pairs. Additionally, we developed a two-pathogen, meta-population mechanistic transmission model to capture the co-epidemic dynamics of IAV and SARS-CoV-2, and to quantify the strength and duration of their bidirectional interactions.

**Results** Among all virus pairs examined, a statistically significant association is identified only between IAV positivity and subsequent SARS-CoV-2 risk. When IAV positive rate percentile is between the 52nd and 88th percentiles, the relative risk (RR) of SARS-CoV-2 infection is significantly reduced. The lowest RR for SARS-CoV-2 (0.58, 95% CrI: 0.40-0.85) occurs at a 5-week lag when IAV positivity reaches the 70th percentile. The fitted mechanistic model using incidence data in Beijing shows that IAV infection substantially reduces infection to SARS-CoV-2 by 94.24% (95% CrI: 88.50%–99.24%), with the protective effect lasting 38.24 days (95% CrI: 35.50–41.29 days). Conversely, SARS-CoV-2 infection is associated with a slight increase in infection to IAV.

**Conclusions** Our findings indicate that IAV circulation may transiently reduce population-level infection to SARS-CoV-2, potential through ecological or immunological mechanisms.

## Plain language summary

This study looks at how three common respiratory viruses - SARS-CoV-2, influenza, and RSV, which cause COVID-19, flu and common colds - affect one another when they spread in communities. We used two complementary approaches: advanced statistical model, which identify patterns in real-world data, and mechanistic transmission model, which simulates how viruses spread from person to person. Together, these methods allowed us to measure how strong these interactions are and how long their effects last. The data came from three years of virus activity across seven countries and regions, providing us a broad view across time and places. We found that increases in flu activity, especially influenza A, may reduce the risk of COVID-19 spread in the weeks that follow. However, these virus interactions are complex. They change over time and depend on how much of each virus is circulating. This means that viruses do not spread in isolation, and one can potentially influence the timing and size of another epidemic. Our study shows why it is important to consider interactions between viruses when forecasting future outbreaks and planning public health interventions, especially since many respiratory viruses tend to circulate at the same time of year.

[1]Beijing Key Laboratory of Surveillance, Early Warning and Pathogen Research on Emerging Infectious Diseases, Beijing Center for Disease Prevention and Control, Beijing, China. [2]Beijing Research Center for Respiratory Infectious Diseases, Beijing, China. [3]School of Public Health, Capital Medical University, Beijing, China. [4]WorldPop, School of Geography and Environmental Science, University of Southampton, Southampton, UK. [5]Department of Population Health Sciences, Virginia Tech, Blacksburg, VA, 24061, USA. [6]These authors contributed equally: Yonghong Liu, Xiaoli Wang. ✉e-mail: Shengjie.Lai@soton.ac.uk; wangqy@bjcdc.org; yangpengcdc@163.com

The prospect of a respiratory co-epidemic—the concurrent circulation of respiratory pathogens such as SARS-CoV-2, influenza virus (both influenza A virus [IAV] and influenza B virus [IBV]), and respiratory syncytial virus (RSV) has become a growing concern for public health systems worldwide[1]. Such co-epidemics could overwhelm healthcare infrastructures due to a surge in cases of respiratory infections. Infection with one pathogen may influence susceptibility, transmissibility, or disease severity to subsequent pathogen infection through mechanisms such as immune activation, ecological niche competition, or behavioral changes, a phenomenon known as pathogen-pathogen interaction[2–5]. These interactions can be asymmetrical, for example, infection with virus A may influence infection to virus B, but not necessarily the reverse. Moreover, these effects may diminish as time goes on. Although these interactions occur at the individual level, they can cumulatively lead to substantial effects at the population level, potentially altering the timing, intensity, and synchrony of epidemics[3]. It is vital to accurately predicting and being prepared for potential co-epidemic outbreaks[6].

Studies have highlighted complex interactions among respiratory viruses such as influenza virus and RSV[3,4]. However, findings from these studies remain inconsistent and sometimes contradictory. For instance, animal studies in mice and ferrets suggest that influenza infection can suppress RSV replication[7,8], a mathematical study using data from Hong Kong and Canada found a moderate to strong negative bidirectional interaction between the two viruses[9]. However, a longitudinal household study in humans in South Africa across all age groups showed the opposite effect, with RSV infection increasing the risk of subsequent influenza infection[10].

This conflicting evidence complicates efforts to predict and mitigate the co-circulations of respiratory viruses, particularly with SARS-CoV-2. While several studies have explored the interaction between influenza virus and SARS-CoV-2, most have been conducted in experimental settings with animal models, limiting their generalizability to human populations. For example, a mouse experiment[11] showed that SARS-CoV-2 infection did not interfere with IAV replication, whereas the replicative ability of SARS-CoV-2 was significantly reduced after IAV infection. Research has also showed that IAV promoted SARS-CoV-2 replication in human cells by upregulating the cell's expression of angiotensin converting enzyme 2, the homologous receptor of SARS-CoV-2[12], but Gilbert-Girard et al.[13] used nasal human airway epitheliums showed that IAV interferes with the replication of SARS-CoV-2, whereas SARS-CoV-2 does not significantly interfere with IAV. In human populations, epidemiological evidence also varies. A test-negative conducted in England from January to April 2020 reported that the risk of testing positive for SARS-CoV-2 among influenza-positive cases was reduced by 58%[14]. In contrast, a population-based SARS-CoV-2 transmission model incorporated influenza, and fitted early pandemic data from Belgium, Italy, Norway, and Spain indicated that influenza increased the infection risk of SARS-CoV-2[15]. Similarly, the interaction between RSV and SARS-CoV-2 remains understudied, with only a cell-based experiment suggesting that RSV infection may inhibit SARS-CoV-2 replication[16].

Our findings show that population-level circulation of IAV is associated with a subsequent reduction in SARS-CoV-2 infection risk, with a transient protective effect that lasts for several weeks. Mechanistic modeling further suggests that this interaction is asymmetric, with prior IAV infection substantially reducing susceptibility to SARS-CoV-2, whereas SARS-CoV-2 infection is linked to only a slight increase in IAV risk. These results highlight the importance of pathogen–pathogen interactions in shaping respiratory virus co-circulation and suggest that such interactions should be considered in the interpretation of surveillance data and in the planning of future prevention and control strategies.

## Methods
### Data sources and collection
Weekly positivity rates of four respiratory viruses, SARS-CoV-2, IAV, IBV, and RSV from October 2021 to May 2024 were obtained from seven different countries/regions in the northern hemisphere: England, Denmark, Ireland, Portugal, Slovenia, the United States (divided into 10 US Department of Health and Human Services [HHS] regions) and Beijing in China. These locations were selected based on data quality, availability, and completeness over the study period (supplementary information). Specifically, they provided openly available high temporal resolution, consistent surveillance data on all four viruses, allowing us to examine viral interactions with minimal data gaps. In the US, we leveraged the 10 HHS regions to account for spatial and temporal heterogeneity within a vast country with large, diverse population. This allowed us to capture regional variations in virus circulation patterns while ensuring sufficient spatial granularity for accurate modeling. England, Denmark, Ireland, Portugal, and Slovenia also offered a broad geographic spread across Europe, enabling the study to assess viral interactions across varied climates, population densities, and healthcare systems. Additionally, these countries demonstrated strong reporting accuracy and relatively minimal disruptions in testing capacity during the COVID-19 pandemic, which was crucial for maintaining data integrity[17].

The pathogen positivity rate was defined as the proportion of positive samples among all the samples tested using Polymerase chain reaction (PCR). Details on the pathogen surveillance for each region are provided in the supplementary information. Given the continuous mutation of SARS-CoV-2 during the study period, we controlled for variant effects by incorporating data on variant composition. The proportion data on SARS-CoV-2 variant composition were derived from the Global Initiative on Sharing Avian Influenza Data (GISAID)[18], categorizing into five dominant lineages/groups: XBB, BA.2.75, BA.2.86, other Omicron variants (excluding XBB, BA.2.75 and BA.2.86), and non-Omicron variants.

Our analysis period began in October 2021, after the relaxation of the most stringent non-pharmaceutical interventions (NPIs) used to control COVID-19, which had also previously suppressed the circulation of other respiratory infections[19–21]. This period allowed for clearer observation of more natural virus transmission patterns and interactions, as NPIs were lifted or gradually reduced across the selected countries. Our analysis also accounted for the varying levels of NPI implementation across different regions during the study period. NPIs data were obtained from the Oxford COVID-19 Coronavirus Government Response Tracker (OxCGRT, https://github.com/OxCGRT/covid-policy-dataset)[22]. OxCGRT stopped collecting data for China by 28 February 2023 and for all other countries at the end of 2022, so we used the R package 'imputeTS' to extrapolate NPI data for 2023 and 2024, employing constant interpolation based on the most recent available values.

To further account for environmental and socioeconomic confounders across regions, we incorporated gross domestic product, population density, and age structure data from the World Bank and national statistical bureaus. We also obtained the Universal Health Coverage (UHC) Service Coverage Index[23] from the World Health Organization (WHO), to represent the extent and quality of health services available to populations for each country (https://data.who.int/indicators/i/3805B1E/9A706FD). The Sociodemographic Index (SDI)[24], a composite indicator of factors including income per capita, average educational attainment and fertility rate, was gathered from the Institute for Health Metrics and Evaluation (https://ghdx.healthdata.org/record/gbd-2023-socio-demographic-index-sdi) to indicate development status. Meteorological data for each country, including weekly average temperature and humidity, were collected from the National Centers for Environmental Information (https://www.ncei.noaa.gov/maps/daily/)[25].

### Estimated SARS-CoV-2 and IAV incidence in Beijing
Since the reported case numbers may underestimate the true incidence of IAV and SARS-CoV-2 infections, we used a multiplicative model to estimate the actual number of infections, as described in Reed et al.[26]. We estimated incidence data from February 2023 to July 2024, after a period during which Beijing had just experienced the first major wave of COVID-19. In preceding years in 2021–2022, IAV had not entered a typical epidemic season, leaving most of the population susceptible to IAV. The model

adjusts the reported case counts by incorporating five parameters: (A) the proportion of infections that are symptomatic, (B) the proportion of symptomatic infections that meet the influenza-like illness (ILI) definition, (C) the healthcare-seeking rate among ILI cases, (D) the success rate of collecting throat swab samples, and (E) the test sensitivity of PCR assays. Parameters values for SARS-CoV-2 and influenza were obtained from Zhang et al.[27] and Wu et al.[28], respectively. The variability and uncertainty of the model parameters were addressed using the Monte Carlo method. For each parameter included in the model, a uniform probability distribution was applied covering the range from the minimum to the maximum value, and the model randomly sampled from these distributions over 10,000 iterations.

### Ethical approval

Ethical clearance for collecting and using secondary data in this study was granted by the institutional review board of the University of Southampton (No. 87924). All data were supplied and analyzed in an anonymous format, without access to personal identifying information.

### Statistical model

We specified a spatiotemporal Bayesian hierarchical model, in which the responses included the weekly positivity rates of viruses for each study area (except Beijing) from October 2021 to May 2024. Due to the variability in surveillance practices across regions, the positivity rate for each region was normalized to a range of 0–100, with logarithmic transformation applied to mitigate skewness. In regions where positivity rates were zero, we added $10^{-3}$ to all values to avoid errors in the log transformation. To examine viral interactions, a DLNM model was included to estimate the exposure-lag-response associations between the virus positivity rates, in terms of relative risks (RRs). The DLNM framework allowed us to assess how current and lagged exposures to one virus affected the spread of another[29]. Control covariates included the Stringency Index, prior immunity levels (defined as the median positivity rate from the past year), SARS-CoV-2 variant composition, and meteorological factors. We also included spatiotemporal random effects to account for potential unobserved and unmeasured sources of heterogeneity and dependencies. The model specification is as follows:

$$\log\left(positivity\ rate_{i,c,t}\right) = NS(positive\ rate_{j,c,t},\ var\ df,\ lag\ df) + NPI_{c,t}$$
$$+ immunity_{i,c,t} + climate\ factor_{c,t} + variant_{x,c,t} + \alpha_{c,t} + \mu_{c,t} + \varphi_{c,t}$$

$$(1)$$

where $positivity\ rate_{i,c,t}$ represents the positivity rate of virus $i$ in country $c$ at week $t$. We included a distributed lag non-linear model formulation using natural cubic spline (NS) in R packages *dlnm* and *splines* to quantify the exposure-lag-response association between virus $i$ and virus $j$. $NPI_{c,t}$ and *climate factor*$_{c,t}$ is the Stringency Index and the mean temperature or relative humidity in country $c$ at week $t$, respectively. $immunity_{i,c,t}$ is the median positivity rate from the past year for virus $i$ in country $c$ at week $t$. For SARS-CoV-2 positivity as responses, $variant_{x,c,t}$ represents the composition of SARS-CoV-2 lineage/group $x$ in country $c$ at week $t$. $\alpha_{c,t}$ is the seasonal random effects for country $c$, a cyclic first order random walk prior was used, which allows each week to depend on the previous week. For the 10 HHS regions in the US, we also considered the internal spatial random effects, we used a modified Besag-York-Mollié model, which consists of one precision parameter and one mixing parameter that determines the relative contribution of spatially structured $\mu_{c,t}$ and unstructured $\varphi_{c,t}$ random effects[30]. For all random effects, we used the penalized complexity prior to the precision t = $1/\sigma^2$, so that $\Pr(1/\sqrt{t} > 0.5) = 0.01$[31].

We trained the model using data from six different countries (US - ten HHS regions, Denmark, Ireland, Portugal, Slovenia, and England). The model parameters are estimated using the integrated nested Laplace approximation in the Bayesian framework in R version 4.3.1. Model selection was guided by the deviance information criterion (DIC) and cross-

validated log score. The DIC balances model accuracy and complexity by penalizing the number of effective parameters in the model, while the mean cross-validated log score evaluates the predictive ability. Therefore, the optimal model minimized both metrics to balance model complexity and predictive performance. We also did a cross-validation, by refitting the selected model, excluding four weeks from the fitting process each time, and compared observations to out-of-sample posterior predictive distributions for each country. Additionally, we used the data from Beijing during the period from December 2022 to May 2024 after the strict COVID-19 NPIs lifted[32], to perform external validation of the model posterior predictive distributions.

To combine specific effects from different locations, when modeling the exposure-lag-response relationship in each region, the exposure factors were standardized to percentiles, ensuring that the range of the exposure factors was consistent and pooled the exposure-response relationship. We then conducted a multivariate meta-regression to explore the pooled exposure-lag-response association across the seven different regions, including Beijing, adjusting for potential region-specific confounders, such as GDP, UHC index, SDI, population density, and age structure. Due to multicollinearity among these variables, we retained only a subset of core predictors with minimal redundancy for the multivariate meta-regression analysis. Akaike information criterion (AIC) was used to measure the goodness of model fits. The residual heterogeneity was measured and tested by the multivariate extension of $I^2$ statistics and Cochran $Q$ test.

To minimize potential dilution of interaction effects during off-season periods, we performed analyses using both the full time series and a subset restricted to the typical epidemic seasons of influenza and RSV (October to March), reflecting the distinct seasonality of these viruses in the Northern Hemisphere[33].

### Mathematical modeling of IAV and SARS-CoV-2 interactions

To further quantify the interaction between IAV and SARS-CoV-2, we developed an 2-pathogen susceptible infected protected recovered (SIPR*SIPR) model based on the available, estimated incidence data in Beijing. In this model, each compartment represents the infection status of an individual for both viruses. For example, the compartment $SI$ indicates that an individual is susceptible to SARS-CoV-2 but infected with influenza. In total, there are 16 possible compartment types representing all combinations of the two pathogens' infection statuses (Supplementary Fig. 1).

The model incorporates two key parameters to capture viral interactions: the interaction strength ($\sigma$) and the duration of interaction ($\rho$). The parameter $\sigma$ has a possible range of $[-1, +\infty)$, where $\sigma = -1$ indicates complete suppression of susceptibility to the subsequent infection, values between $-1$ and $0$ indicate partial reduction in susceptibility, $\sigma = 0$ represents no effect, and $\sigma > 0$ indicates increased susceptibility. The parameter $\rho$ denotes the duration of the interaction effect.

Prior distributions for $\sigma$ and $\rho$ were informed by results from statistical analyses. Posterior estimates were obtained by fitting the model to real-world data from Beijing. Given that the coverage of both influenza and SARS-CoV-2 vaccination has been very low in Beijing, influenza vaccine coverage at approximately 20%[34] and vaccination rates against the Omicron variant of SARS-CoV-2 remaining very low, the impact of vaccination was not considered in the model. The full model is as follows:

$$\frac{dSS}{dt} = -\frac{\beta_1 SS(IS + II + IP + IR)}{N} - \frac{\beta_2 SS(SI + II + PI + RI)}{N} + \omega_1 RS + \omega_2 SR$$

$$(2)$$

$$\frac{dIS}{dt} = \frac{\beta_1 SS(IS + II + IP + IR)}{N} - \frac{\beta_2(1 + \sigma_1)IS(SI + II + PI + RI)}{N} - \gamma_1 IS + \omega_2 IR$$

$$(3)$$

$$\frac{dPS}{dt} = \gamma_1 IS - \rho_1 PS - \frac{\beta_2(1 + \sigma_1)PS(SI + II + PI + RI)}{N} + \omega_2 PR \quad (4)$$

$$\frac{dRS}{dt} = \rho_1 PS - \frac{\beta_2 RS(SI + II + PI + RI)}{N} - \omega_1 R + \omega_2 RR \quad (5)$$

$$\frac{dSI}{dt} = \frac{\beta_2 SS(SI + II + PI + RI)}{N} - \frac{\beta_1(1 + \sigma_2)SI(IS + II + IP + IR)}{N} - \gamma_2 SI + \omega_1 RI \quad (6)$$

$$\frac{dII}{dt} = \frac{\beta_1(1 + \sigma_2)SI(IS + II + IP + IR)}{N} + \frac{\beta_2(1 + \sigma_1)IS(SI + II + PI + RI)}{N} - \gamma_1 II - \gamma_2 II \quad (7)$$

$$\frac{dPI}{dt} = \frac{\beta_2(1 + \sigma_1)PS(SI + II + PI + RI)}{N} + \gamma_1 II - \gamma_2 PI - \rho_1 PI \quad (8)$$

$$\frac{dRI}{dt} = \frac{\beta_2 RS(SI + II + PI + RI)}{N} + \rho_1 PI - \gamma_2 RI - \omega_1 RI \quad (9)$$

$$\frac{dSP}{dt} = \gamma_2 SI - \rho_2 SP - \frac{\beta_1(1 + \sigma_2)SP(IS + II + IP + IR)}{N} + \omega_1 RP \quad (10)$$

$$\frac{dIP}{dt} = \gamma_2 II + \frac{\beta_1(1 + \sigma_2)SP(IS + II + IP + IR)}{N} - \gamma_1 IP - \rho_2 IP \quad (11)$$

$$\frac{dPP}{dt} = \gamma_1 IP + \gamma_2 PI - \rho_1 PP - \rho_2 PP \quad (12)$$

$$\frac{dRP}{dt} = \gamma_2 RI + \rho_1 PP - \rho_2 RP - \omega_1 RP \quad (13)$$

$$\frac{dSR}{dt} = \rho_2 SP - \frac{\beta_1 SR(IS + II + IP + IR)}{N} - \omega_2 SR + \omega_1 RR \quad (14)$$

$$\frac{dIR}{dt} = \frac{\beta_1 SR(IS + II + IP + IR)}{N} + \rho_2 IP - \gamma_1 IR - \omega_2 IR \quad (15)$$

$$\frac{dPR}{dt} = \gamma_1 IR + \rho_2 PP - \rho_1 PR - \omega_2 PR \quad (16)$$

$$\frac{dRR}{dt} = \rho_1 PR + \rho_2 RP - \omega_1 RR - \omega_2 RR \quad (17)$$

$$\beta_1 = B_1\left(1 + c_1 \cos\left(\frac{2\pi t}{365/2}\right) + d_1\right) \quad (18)$$

$$\beta_2 = B_2\left(1 + c_2 \cos\left(\frac{2\pi t}{365}\right) + d_2\right) \quad (19)$$

where {1,2} denotes the infection status of individuals with respect to SARS-CoV-2 and IAV, with S for susceptible, I for infectious refractory phase, P for noninfectious refractory phase, R for immunity phase. The definitions and values of parameters in the model are provided in Table 1. The initial values for the model compartments, IS, RS, SI, SR, and RR, were treated as parameters to be estimated, while all other compartments were initialized at zero.

Based on the estimated incidence data in Beijing from February 2023 to July 2024, model fitting was performed using the Metropolis–Hastings Markov chain Monte Carlo (MCMC) algorithm with the MATLAB (version R2022a) toolbox DRAM (Delayed Rejection Adaptive Metropolis). To account for seasonal variation in transmissibility, certain model parameters, such as the transmission rates of each virus, were assumed to vary seasonally. A segmented fitting approach was adopted to estimate these parameters as well as the interaction effects between the viruses. Given that the epidemic cycle of SARS-CoV-2 in Beijing averaged approximately six months (Supplementary Fig. 11), the model fitting was conducted in two stages.

In the first stage, the model was fitted in segments of 180 days each to estimate parameters related to seasonal variation, including the amplitude and phase of seasonal forcing for each virus. These seasonal parameters, which describe the periodic changes in transmissibility, were assumed to remain constant throughout the entire study period. In the second stage, the model was fitted to the full time series to estimate the parameters governing viral interactions, specifically the interaction strength ($\sigma$) and the duration of interaction ($\rho$). In this stage, the seasonal parameters for SARS-CoV-2 were fixed to the values estimated in the first-stage segmented fitting, reflecting changes in transmissibility due to ongoing variant replacements. For influenza, which exhibits more stable seasonality, the seasonal parameters estimated from the first stage were used as priors, and the transmission rate was re-estimated over the entire period. The parameters describing viral interactions were also re-estimated in this stage.

After a burn-in of 500,000 iterations, we ran the MCMC simulation for a further 1 million iterations, sampled at every 1000th step to avoid autocorrelation. Trace plots and Gelman and Rubin diagnostics were used to judge convergence of the MCMC chains. Each fitting exercise was repeated three times to test the robustness of results, which converged to the same estimates on each occasion (Supplementary Fig. 12).

## Statistics and reproducibility

The analyses were conducted using R (version 4.3.1) and MATLAB (version R2022a). Detailed descriptions of the statistical methods and the availability of code and data are provided in the manuscript. All results were reproducible under the procedures described.

## Results

### Seasonality

From October 2021 to May 2024, IAV and RSV showed overlapping seasonal dynamics across study locations and countries. Both viruses exhibited clear seasonal patterns, peaking during winter or spring, with RSV generally reaching its peak earlier than IAV (Fig. 1). The weekly positivity rates of IAV ranged approximately from 0% to 67.5%, while RSV ranged from 0% to 32.6% across all regions and seasons. In contrast, the positivity rate of IBV maintained low (ranging from 0% to 14.7%) throughout the study period, except during the 2023/2024 epidemic season in Beijing, leading to its exclusion from the subsequent modelling of viral interactions. Unlike IAV and RSV, SARS-CoV-2 displayed no distinct seasonal patterns, with irregular timing and magnitude of waves due to the emerging variants[36]. Its weekly positivity rate fluctuated between 0.4% and 57.2%, varying greatly by region and time. The epidemic patterns of these viruses were synchronized across the 10 HHS regions in the US (Fig. 1A), whereas significant differences were observed across the other six countries (Fig. 1B, C).

### Interactions between viruses

We used DLNMs within a Bayesian modelling framework. The inclusion of relative humidity, NPI, and immunity resulted in a greater reduction in the DIC and mean logarithmic score compared with the baseline model (Supplementary Tables 1–18). We found a significant negative interaction was revealed between IAV infection and subsequent SARS-CoV-2 risk when IAV positivity was between 64th percentile and 80th percentile (Fig. 2A–C). The risk of subsequent SARS-CoV-2 infection reached a minimum RR of 0.43 (95% CrI: 0.21–0.85) at the 70th percentile (Fig. 2C). However, this protective effect diminished as IAV positivity increased beyond the 80th percentile. No significant association was found between SARS-CoV-2 positivity and subsequent risk of IAV infection (Supplementary Fig. 2A–C). For the IAV/RSV pair, a nonlinear positive interaction was observed (Fig. 2D–I). The highest risk of IAV infection occurred when RSV positivity was at the 68th percentile (RR 11.08, 95% CrI: 5.12–23.98), while the risk of RSV infection peaked at an IAV positivity of the 46th percentile (RR 7.25, 95% CrI: 2.97–17.70). For the SARS-CoV-2/RSV pair, no significant bidirectional associations were observed (Supplementary Fig. 2D–I).

**Table 1 | The definitions and values of parameters in the mathematical model**

| Parameter | Description | Values (95% CI) | Source |
|---|---|---|---|
| $\sigma_1$ | Strength of the interaction effect of SARS-CoV-2 on IAV | -0.08 (-0.01, 0.17) | Fitted |
| $\sigma_2$ | Strength of the interaction effect of IAV on SARS-CoV-2 | -0.94 (-0.99, -0.89) | Fitted |
| $1/\rho_1$ Rate of loss of cross-protection against IAV after SARS-CoV-2 infection | 3.23 (1.13, 6.30) | Fitted | |
| $1/\rho_2$ Rate of loss of cross-protection against SARS-CoV-2 after IAV infection | 38.24 (35.50, 41.29) | Fitted | |
| IS | Initial number of the population infected with SARS-CoV-2 only | 1003.4 (922.06, 1091.7) | Fitted |
| RS | Initial number of the population immune to SARS-CoV-2 only | 1.68e + 07 (1.6799e + 07, 1.6801e + 07) | Fitted |
| SI | Initial number of the population infected with IAV only | 3009.5 (2919.5, 3093.4) | Fitted |
| SR | Initial number of the population immune to IAV only | 408.94 (310.83, 498.52) | Fitted |
| RR | Initial number of the population immune to both SARS-CoV-2 and IAV | 991.19 (897.81, 1084.4) | Fitted |
| $B_2$ | Rate of IAV transmission | 1.15 (1.14, 1.16) | Fitted |
| $c_2$ | Seasonal amplitude of IAV forcing parameter | 0.08 (0.08, 0.09) | Fitted |
| $d_2$ | The week during which the force of infection is maximal | −8.01 (−8.06, −7.96) | Fitted |
| $1/\gamma_1$ | Infectious period of SARS-CoV-2 | 5 days | Hakki et al.[44] |
| $1/\gamma_2$ | Infectious period of IAV | 5 days | Lei et al.[45] |
| $\omega_1$ | Rate of waning immunity against SARS-CoV-2 | 1/180 | Chen et al.[46] |
| $\omega_2$ | Rate of waning immunity against IAV | 1/(2*365) | Lei et al.[45] |

We conducted both internal and external cross-validation on the significant interaction models across seven regions. We applied a rolling block cross-validation approach, iteratively holding out 4-week segments to generate posterior predictive (Supplementary Figs. 3–9). External validation was performed using surveillance data from Beijing (Supplementary Fig. 9). The results indicated that the model demonstrated strong predictive performance in both internal and external validations, accurately identifying epidemic outbreaks, with the exception of cases where the positivity rate was 0%, where the model's performance was poor.

Due to the distinct seasonality of influenza and RSV, with high incidence in winter and low incidence in summer, we further excluded non-epidemic season data from the statistical modelling. Among all virus pairs, a significant association was observed only between IAV positivity and subsequent SARS-CoV-2 risk (Fig. 3 and Supplementary Fig. 10). When IAV rate was between the 52nd and 88th percentiles, the RR of SARS-CoV-2 was significantly reduced (Fig. 3B). The minimum RR for SARS-CoV-2 was 0.58 (95% CrI: 0.40–0.85) at a 5 week-lag when IAV positivity was at the 70th percentile (Fig. 3D). By incorporating population density, SDI and age structure into the model, we could decrease the residual heterogeneity to 87.5% ($p < 0.001$) (Supplementary Table 19).

**Co-transmission pattern shaped by viral interaction**
Our statistical modelling results indicated a significant negative association between percentile of IAV positivity and subsequent risk of SARS-CoV-2 infection, both when using the full dataset and when restricting to the influenza epidemic season. To further quantify the direction, strength, and duration of the interaction between IAV and SARS-CoV-2, we constructed a two-pathogen, meta-population mechanistic transmission model incorporating an interaction term. This model was fitted to observed infections of IAV and SARS-CoV-2 in Beijing (Fig. 4). Model fitting revealed that IAV infection substantially reduced infection to SARS-CoV-2 by 94.24% (95% CrI: 88.50–99.24%), with the protective effect lasting approximately 38.24 days (95% CrI: 35.50–41.29 days). Conversely, SARS-CoV-2 infection was associated with a slight increase of 7.91% in infection to IAV (95% CrI:

−1.34% to 16.87%), with a duration of protection of around 3.23 days (95% CrI: 1.13–6.30 days). The results of the segmented fitting are shown in Supplementary Fig. 12.

To assess the impact of viral interaction on the co-circulation patterns of SARS-CoV-2 and IAV, we simulated the epidemic trajectories under a counterfactual scenario in which IAV exerts no negative interaction on SARS-CoV-2 (Supplementary Fig. 13). We found that the COVID-19 epidemic peaks would occur earlier and be substantially higher in magnitude. Specifically, the summer 2023 peak would advance by approximately two weeks, with a 2.35-fold (range: 1.67–3.15) increase in peak magnitude, while the winter 2024 peak would advance by about six weeks, with a 3.52-fold (range: 2.21–5.09) increase in magnitude. Overall, our findings suggest that IAV may suppress SARS-CoV-2 transmission, delaying its epidemic progression and attenuating peak intensity, thereby reducing the burden on healthcare resources.

**Discussion**
In this study, we investigated the nonlinear interactions and time-lag effects of co-epidemics between three major respiratory viruses—SARS-CoV-2, influenza, and RSV—at the population level, which provides valuable insights for informing future public health strategies and preparedness. We found that prior influenza infection could significantly reduce the risk of subsequent SARS-CoV-2 infection, with the lowest risk of SARS-CoV-2 occurring approximately one month after IAV positivity reached the 70th percentile. This finding aligns with the results of experiments in mice[11] and nasal human airway epitheliums[13], which showed that SARS-CoV-2 replication was significantly reduced following IAV infection. This phenomenon can be attributed to the heightened sensitivity of SARS-CoV-2 to interferon (IFN)[37], which is produced by the antiviral response triggered by IAV infection[16].

Our DLNM results suggest an association between increased population-level IAV activity and a reduced risk of SARS-CoV-2 infection, with a delay of approximately 4–6 weeks. Two potential mechanisms could explain this lag effect: first, immune interference, where the immune

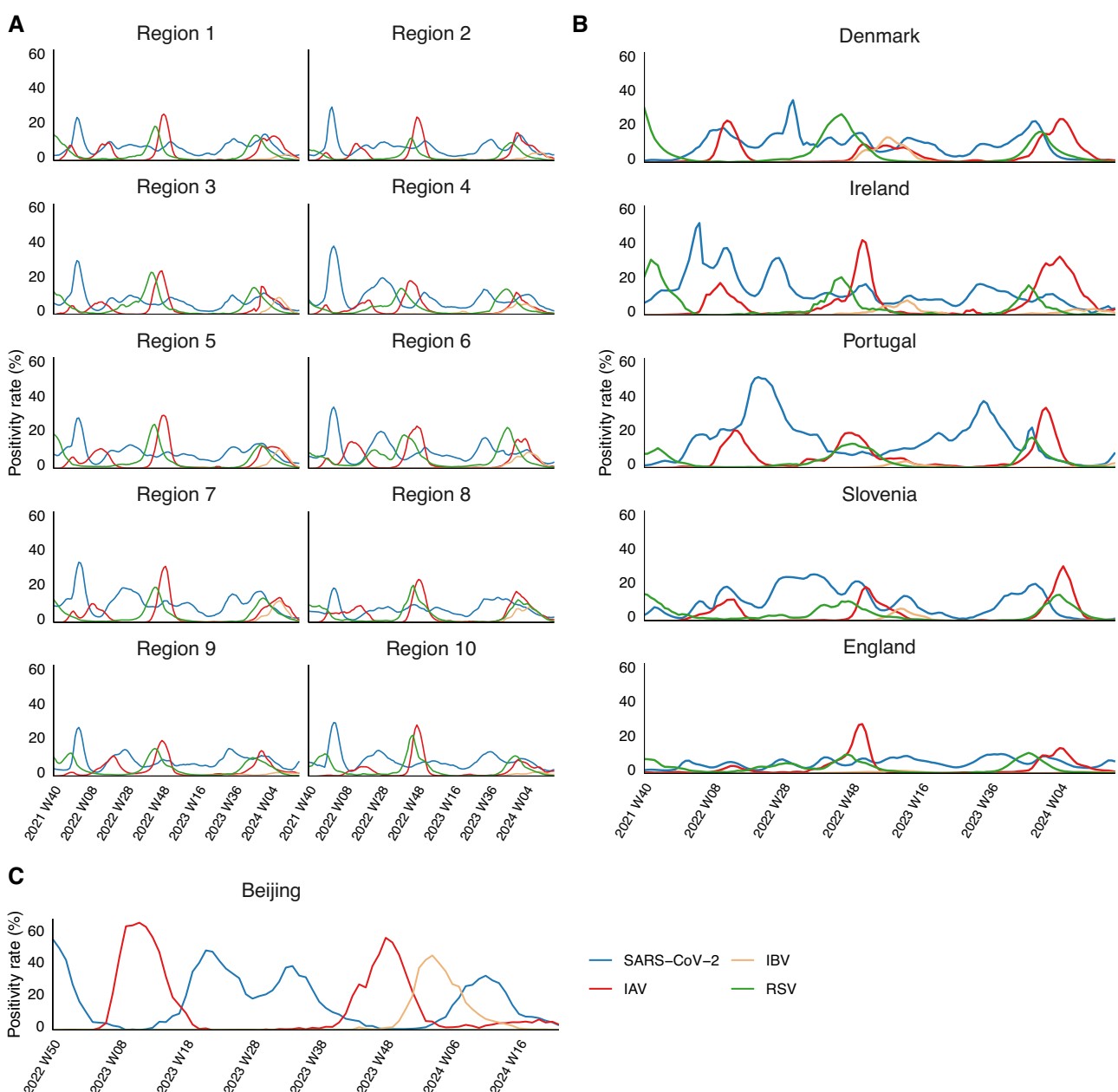

**Fig. 1 | Weekly positivity rates of four respiratory viruses in different study locations and countries from October 2021 to May 2024.** Blue represents SARS-CoV-2, red represents IAV, yellow represents IBV, green represents RSV. **A** Ten United States HHS regions[35]. **B** Denmark, Ireland, Portugal, Slovenia, and England. **C** Beijing, China.

response in the host to one virus reduced the replication of another[8], leading to a suppressed transmission during the epidemic peak of the former[3]. Park et al.[2] found infection with one pathogen at the individual level may reduce the susceptible population at the community level due to niche competition, thereby altering the overall transmission pattern of another pathogen. This ecological interaction can result in replacement or delayed epidemic effects at the population scale. A similar phenomenon was observed in France during the 2009 H1N1 pandemic, where widespread rhinovirus circulation in the autumn significantly delayed the onset of H1N1 influenza by approximately four weeks[38]. Second, behavioral factors, as the onset of flu season, may increase awareness and adherence to protective measures, indirectly reducing SARS-CoV-2 transmission. Notably, no significant effect of prior SARS-CoV-2 infection on influenza risk was observed, consistent with previous research indicating minimal cross-virus interference[11].

To further investigate the interaction between SARS-CoV-2 and IAV, we developed a two-pathogen transmission dynamic model and fitted observed infection in Beijing. The model revealed that IAV infection

significantly suppressed the risk of SARS-CoV-2 infection, with an estimated refractory period of approximately 30 days. To our knowledge, this is the first study to quantify the interaction between IAV and SARS-CoV-2 using a two-pathogen transmission dynamic modeling framework calibrated with observed infection of population-level data. By explicitly capturing the co-epidemic transmission patterns of IAV and SARS-CoV-2, this model provides robust support for the hypothesis of viral interference at the population level. From a public health perspective, incorporating pathogen–pathogen interactions into epidemic models could improve short-term forecasting during respiratory virus seasons and inform better preparedness strategies. For instance, early influenza activity may offer a temporary window of reduced susceptibility to SARS-CoV-2, which could influence the optimal timing of intervention measures such as vaccination campaigns and healthcare resource allocation.

In contrast, we identified a weak positive association between IAV and RSV. The risk of IAV infection was highest when RSV positivity was at the 68th percentile, and the risk of RSV infection peaked when IAV positivity

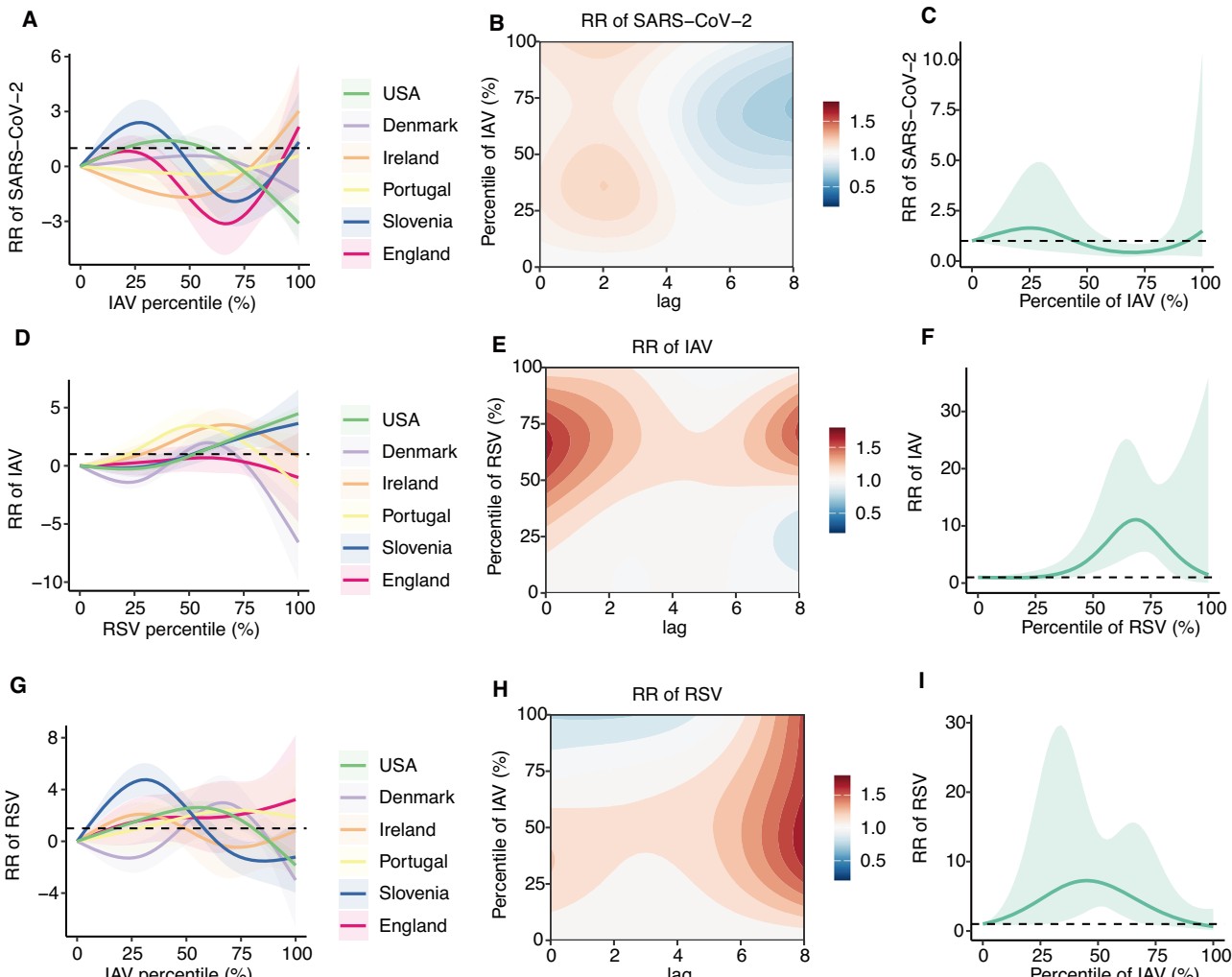

**Fig. 2 | Exposure-response association between viruses.** Associations between the percentile of IAV positivity and subsequent SARS-CoV-2 infection risk. **A** RR of SARS-CoV-2 associated with different percentiles of IAV positivity across regions, shaded areas representing 95% CrI. **B** Pooled contour plot of the lag–response association between percentile of IAV positivity and SARS-CoV-2 risk, relative to the minimum IAV percentile. **C** Pooled exposure–response association between

percentile of IAV positivity and SARS-CoV-2 risk, shaded areas representing 95% CrI. **D–F** Associations between percentile of RSV positivity and subsequent IAV infection risk. **G–I** Associations between percentile of IAV positivity and subsequent RSV infection risk. Estimates are based on n = 6168 biologically independent weekly observations included in the DLNM.

was at the 46th percentile. This finding differs from some experimental studies that suggest suppression of RSV by IAV[7,8], though other research indicates that RSV may increase susceptibility to IAV[10]. RSV's non-structural proteins NS1 and NS2 might contribute to this increased risk by interfering with the host immune response[39,40]. However, the mechanism underlying the interaction between IAV and RSV remains controversial. Age-specific data and analysis are essential for further understanding these dynamics, as RSV primarily affects infants and young children[41], whereas IAV affects individuals of all ages.

Our study shows no statistical evidence of interaction between RSV and SARS-CoV-2, which is consistent with previous experimental results in human nasal epithelial cells. These studies showed that while influenza A(H1N1)pdm09 virus interfered with SARS-CoV-2 replication, RSV did not have a significant impact[42]. Although our findings are supported by experimental data, there is still limited epidemiological or mathematical evidence to confirm this lack of interaction, highlighting the need for further research.

However, our study has several limitations. First, differences in the population coverage of positivity rate testing across countries might introduce variability. In Beijing, positivity rates were based on ILI cases, while in other areas, they were derived from symptomatic patients seeking care, not

necessarily with ILI. To reduce this heterogeneity, we standardized positivity rates within each region and avoided direct inter-regional comparisons. Second, due to the relatively short duration of the endemicity of COVID-19, the study period covered only three epidemic seasons, limiting the long-term validity of our findings. Longer-term data will be required in future research to validate our findings, together with additional experimental and mathematical modeling studies. Third, although we attempted to include more countries and pathogens in the analysis, the availability of reliable pathogen data was limited in several regions, which might result in wider confidence intervals in the meta-analysis. Fourth, we were unable to obtain age-specific data on COVID-19, influenza, and RSV infections in these regions, which could provide more granular insights into viral interactions and transmission patterns. Lastly, since the NPI data stopped updating at the end of 2022, we used a simple forecasting algorithm to extrapolate data for 2023 and 2024, which might affect the reliability of the DLNMs results. Nonetheless, given the full relaxation of control measures in most regions since the second half of 2022, the potential impact of NPIs on our findings is likely minimal.

Despite these limitations, our study provides important evidence of population-level interactions between SARS-CoV-2, influenza, and RSV. These findings can inform the development of more targeted strategies for

**Fig. 3 | Exposure-lag-response association between IAV and SARS-CoV-2 during epidemic seasons. A** RR of SARS-CoV-2 associated with different percentiles of IAV positivity across regions, shaded areas representing 95% CrI. **B** Pooled exposure–response association between percentiles of IAV positivity and SARS-CoV-2 risk, shaded areas representing 95% CrI. **C** Pooled contour plot of the lag–response association between percentiles of IAV positivity and SARS-CoV-2 risk, relative to the minimum IAV percentile. **D** RR of SARS-CoV-2 lag-response association for 70th percentile of IAV positivity, shaded areas representing 95% CrI. Estimates are based on n = 1170 biologically independent weekly observations included in the DLNM.

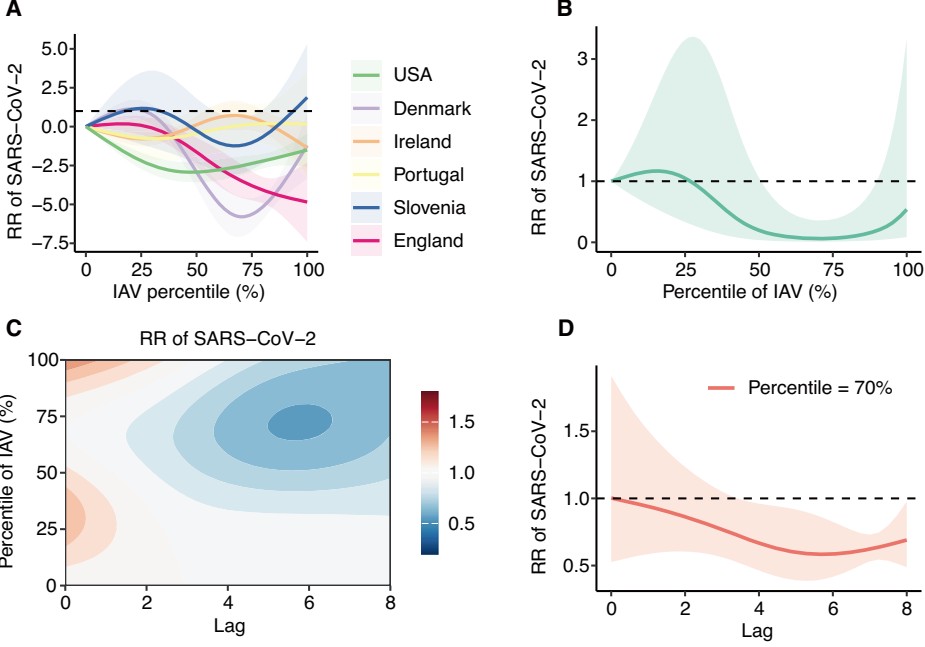

**Fig. 4 | Fitted transmission dynamic model with interactions between IAV and SARS-CoV-2 in Beijing. A** IAV. **B** SARS-CoV-2. Black lines represent the observed incidence, red lines represent the estimated incidence of IAV, blue lines represent the estimated incidence of SARS-CoV-2. Shaded areas representing 95% prediction interval, model was based on n = 541 biologically independent daily surveillance observations.

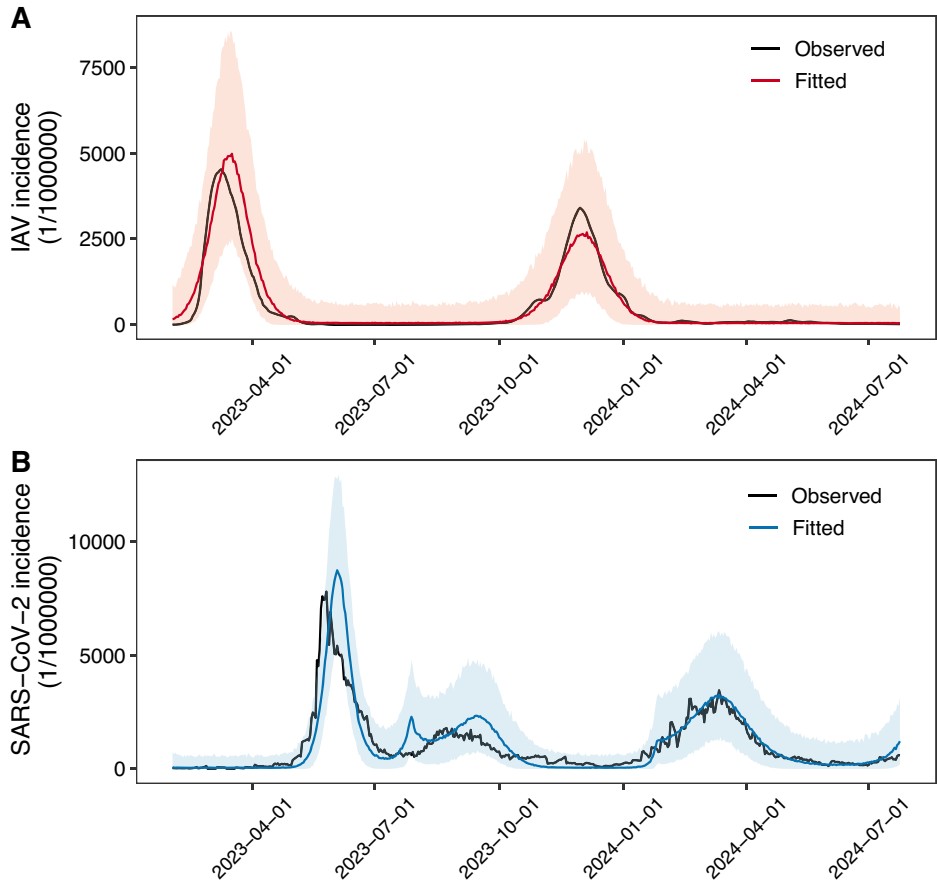

managing respiratory infections. For example, when predicting short-term trends of pathogen circulation, the influence of interacting pathogens should be taken into account to improve forecasting accuracy. Similarly, vaccination strategies may benefit from incorporating knowledge of viral interactions to support coordinated prevention of multiple pathogens and

optimize the effectiveness of immunization programs. To strengthen public health preparedness, future transmission risk assessments and intervention strategies should explicitly consider the potential impact of pathogen interactions and co-epidemics. Further research using long-term, community-level data across multiple pathogens and geographic locations

is also needed to validate these findings and examine the robustness of the underlying assumptions.

## Conclusion

Our study shows that circulation of IAV transiently reduces the risk of SARS-CoV-2 infection at the population level, likely through ecological or immunological mechanisms. The protective effect of IAV is strongest at intermediate levels of IAV activity and lasts several weeks. These findings highlight that respiratory viruses interact in complex ways, and considering such interactions is important for understanding co-epidemic dynamics and informing public health interventions.

## Data availability

The sources of data used in this study can be found in the methods section. The data underlying main Figs. 1A, B, 2–3 and Table 1 can be accessed at https://github.com/yonghongliu02/Virus-virus-interaction[43]. COVID-19 and influenza surveillance data (Figs. 1C and 4) were obtained from the Beijing CDC under a controlled-access agreement. These data were provided in an anonymized and aggregated format without personal identifiers and are subject to data access restrictions. The data in Beijing can be made available upon request to the corresponding authors from Beijing CDC.

## Code availability

The code used to produce the analysis is available at https://github.com/yonghongliu02/Virus-virus-interaction[43].

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

## Acknowledgements

We thank the organizations and staff in different regions for collecting and sharing the data used in this research. We also thank the staff of the 39 sentinel hospitals composed of the Respiratory Pathogen Surveillance System in Beijing for their assistance in respiratory pathogen surveillance. This study was supported by National Natural Science Foundation of China (82504468), the National Institutes of Health (R01AI160780), the U.S. National Science Foundation (DMS-2327797), the Beijing High-Level Innovation and Entrepreneurship Talent Support Program-leading talent projects (G202513070), Special Research Project for Capital Health Development (2024-2G-3016), Key Research Projects of Beijing Natural Science Foundation-Haidian District Joint Fund (L242053), and Research Center for Respiratory Infectious Diseases Project (BJRID2024-014).

## Author contributions

S.L., P.Y., Y.L., and X.W. designed the study. Y.L., M.L., and Y.S. collected the data and did a literature search. Y.L., S.L., M.L., X.W. and E.C. contributed to the methodology. Y.L. and M.L. did the data analysis and visualization. YL wrote the first draft of the manuscript. Q.W., X.W., S.L., E.C., Z.C., W.Z., J.H., H.Y., N.W.R., and A.T. provided important comments on the draft manuscript. All authors reviewed and approved the final draft for submission.

## Competing interests

The authors declare no competing interests.
