## [Transparent Peer Review file · Communications Medicine]

Interactions of SARS-CoV-2, influenza and respiratory syncytial virus influence epidemic timing and risk

Corresponding Author: Dr Shengjie Lai

Version 0:

Reviewer comments:

Reviewer #1

(Remarks to the Author)

1: Overall

In this study, the authors aimed to quantify the interactions of SARS-CoV-2, influenza, and RSV at the population level and their impact on transmission dynamics. They used advanced statistical models to assess the exposure-lag-response associations between these respiratory pathogens. Additionally, they simulated the impact of viral interactions and R_e on co-epidemics. The authors found that Influenza A reduces the risk of SARS-CoV-2 infection, while Influenza A and RSV interact to enhance their epidemic peaks.

2: Comments

Introduction

- Overall, the introduction is somewhat lengthy. The last paragraph can be shortened to a few sentences since the details will be explained in the methods section.
- The third paragraph lacks a clear focus. It describes different studies but does not clearly outline the study design, the population under study, the location, or the time period within the pandemic.
- Is this the first population-based study aiming to assess interactions between different viruses and the impact of co-epidemic dynamics? This approach seems quite different from the studies referenced in the introduction, so a more explicit introduction to this concept is needed. Specifically, how interactions could shape the course of future respiratory outbreaks appears somewhat abrupt.
- Line 73: PIT theory—please write this abbreviation in full.
- Lines 86-92: This sentence is very long and difficult to read. Please split it into 2-3 sentences.
- Line 88: Was the data from Hong Kong and Canada, or was the study conducted there? This needs clarification.
- Line 89: A household study is always in humans. What was the study population—specific households, or only those with children?
- Line 94: Is SARS-CoV-2 still considered "novel"? Consider revising.
- Lines 130 and 145: What does "additional file" refer to? Does this mean supplementary information?
- Supplementary information, Line 27, and Manuscript, Line 126: The main text states that data is from seven different countries/regions, while the supplementary information mentions six. Please clarify or correct this inconsistency.

Methods

- Please provide references for the data sources used, if available.
- Lines 149-151: Are the SARS-CoV-2 variants defined by date, or was strain variant information available in the dataset?
- Is there information available on SARS-CoV-2 and influenza vaccination coverage/uptake? Vaccination status might impact interactions between the viruses and epidemic dynamics.

Results

Seasonality

- What are the positivity rates found for the different viruses? Please provide some numerical values (ranges) in the text, as it is difficult to extract this from the figures.

Interactions between Viruses

- The methods state that analyses were restricted to influenza and RSV epidemic seasons. However, in Lines 275-284, results are presented including all data. I suggest removing this part.

- Line 279: Please refer to supplementary materials in a consistent manner.
- There appears to be no reference to Fig. S2 in the manuscript.
- There appears to be no reference to Fig. 3A, 3D, and 3G in the manuscript.

Co-transmission Pattern Shaped by Viral Interaction and Re

- Line 345: "Which also has been observed in some countries"—this statement should be included in the discussion rather than the results.

Discussion

- Line 373: "Our DLNMs further suggest that the strongest suppressive effect on SARS-CoV-2 occurred 4-6 weeks after an IAV infection." I do not fully understand how to interpret this. Do the results not reflect a decreased risk of SARS-CoV-2 infection as influenza incidence increases, with a delay of 4-6 weeks? As it is currently written, it appears that the data is interpreted as if there is a secondary infection (co-infection) within an individual, whereas the data is at the population level. Please clarify.
- The specific time span of the lower risk of SARS-CoV-2 after influenza infection requires further discussion. The manuscript references experiments in mice, but a brief look at the paper it seems that the mice only have been observed for 14 days. How do you explain that this effect is visible 4-6 weeks after an initial influenza infection and not immediately? I would expect to see this effect in the early period after primary infection due to an initial immune response boost. This comment is also related to the previous one.
- Line 376: The reference from 1959 is quite outdated. Consider citing a more recent study.
- Lines 402-410: Is there additional literature that supports your findings?
- Lines 408-410: How can policymakers account for these interactions? Please elaborate.
- Line 413: Variability in what? This needs clarification.
- Lines 433-434: "For example, during IAV peaks, there may be reduced pressure on SARS-CoV-2 control efforts." This statement seems somewhat outdated, as there are currently no SARS-CoV-2 control measures in place.

Final Remarks:

This is an interesting study that provides valuable insights into interactions between SARS-CoV-2, influenza, and RSV at the population level. However, the manuscript requires some clarifications and structural improvements. Additionally, a more detailed discussion on the timing of the observed effects and their biological plausibility is needed.

Reviewer #2

(Remarks to the Author)

Thank you for the opportunity to review this manuscript. This study investigates the population-level interactions among SARS-CoV-2, influenza A and B (IAV and IBV), and respiratory syncytial virus (RSV) using surveillance data from seven locations between October 2021 and May 2024. By employing distributed lag nonlinear models within a Bayesian spatiotemporal framework and a mechanistic transmission model, the authors quantified the effects of viral co-circulation on disease dynamics. Their findings indicate that IAV reduces the subsequent risk of SARS-CoV-2 infection, while RSV and IAV exhibit a bidirectional positive interaction, mutually amplifying their epidemic peaks.

However, I am not fully convinced that the conclusions are adequately supported by the results.

Major Comments:

1. Figures 2 and 3: I am not fully convinced that the results demonstrate a strong interaction between SARS-CoV-2, influenza, and RSV. The nonlinear changes across seasons and variations between countries suggest that other factors may be influencing the observed patterns. Given that seasonality varies across geographic locations, further clarification is needed on how these differences impact viral interactions. The study also identifies a bidirectional positive interaction between IAV and RSV, but the underlying mechanisms remain unclear—whether due to immune-mediated effects, shared risk factors, or co-infection dynamics.
2. What are the definitions of non-epidemic seasons and epidemic seasons? The exclusion of non-epidemic season data to the statistical analysis raises questions about how this decision impacts the interpretation of viral interactions. It is unclear whether sensitivity analyses were conducted to assess the effect of including non-epidemic season data and how the epidemic season was defined for each virus.
3. The observed asymmetrical interaction between SARS-CoV-2 and IAV suggests that increasing IAV positivity reduces the risk of SARS-CoV-2 infection, yet no evidence was found linking SARS-CoV-2 positivity to subsequent IAV infection. This raises the question of whether external factors, such as vaccination campaigns or public health interventions, influenced these dynamics. Additionally, potential confounders, including variations in testing rates and reporting delays, should be considered.
4. I would suggest that the authors discuss why the nonlinear relationship is observed. For example, the lowest risk of SARS-CoV-2 occurs approximately one month after IAV positivity reaches the 75th percentile. I was wondering what factors contribute to this nonlinear change—specifically, why the lowest risk is achieved after IAV positivity reaches the 75th percentile. How is this related to experimental evidence?
5. Note that an out-of-season outbreak of RSV was observed in the US during the 2021/22 season. How does this influence model results?
6. Figure 4: The behaviour of the interactive transmission model is dependent on population structure, yet it is unclear how the model incorporates differences across geographic locations. While the study employs a meta-population framework, it does not specify whether factors such as age distribution, population density, contact pattern. Additionally, the model assumes fixed reproduction numbers (R_e) and immunity durations for each virus, but it is not evident whether these parameters are adapted to reflect regional epidemiological differences. Given that viral transmission dynamics can vary significantly based on local factors such as vaccination coverage and prior immunity, further clarification is needed on how

the model accounts for these variations. Sensitivity analyses assessing the impact of differing population structures on the model's predictions would strengthen the validity of its findings. Moreover, it would be helpful to understand whether the model outputs have been validated against real-world epidemiological data from the studied locations.

Reviewer #3

(Remarks to the Author)

This manuscript investigates the extent and impact of virus-virus interactions among SARS-CoV2 (SC2), Influenza A virus (IAV) and RSV, using data from multiple locations. In the first part, the authors employ a Bayesian statistical model to estimate interactions among pathogens. In the second part, they define a mechanistic model and use it to show the impact of interactions on epidemic dynamics. The topic of pathogen interactions is an interesting one, but is also riddled with challenges. I believe that this manuscript is well written and suitable for this journal; however, I do have a few concerns that I hope the authors will take into account when revising the article. Please find my comments below:

MAJOR COMMENTS

- My main concern is about the connection between the statistical analysis and the simulation study. As it stands, the latter attempts to recapitulate the impact of statistically significant results found in the former. However, while the motivation for the simulation exercise is justified, the link does not feel sufficiently strong in my opinion. For example, the statistical analysis suggests non-linear relationships (lag- and positivity-wise) between different pairs of viruses; however, it remains unclear whether simulations actually reproduce such suppression/enhancing effects and the lagged dependency with positivity rates. Fig. 2 seems to suggest that pathogens peak at different times when they are not interacting (by the way, I could not find the values of peak times): could such hard-coded shift in seasonality be misunderstood for a displacement induced by virus-virus interactions? What would be the outcome of the DLNM model if applied to simulated data? Would it be able to recover the absence of interactions ($\sigma=0$) and the non-linear dependency with lagged positivity? Finally, it is not entirely clear to me whether the dashed line in Fig. 2 means that the dynamics of one virus is being kept "fixed": I would expect it to change with σ , at least when virus-virus interactions act on both viruses at the same time.
- Fig. 2 shows significant variations among countries, which makes me less confident about some findings. For example, Fig. 2E-F do not seem to suggest a consistent relationship between IAV and RSV.
- In general, I believe that detecting/measuring pathogen-pathogen interactions is extremely difficult and not enough attention is being given to such challenges. I strongly recommend the authors to review relevant work in this area: Shrestha et al, 2011, <https://pubmed.ncbi.nlm.nih.gov/21876665/>; Shrestha et al, 2013, <https://pubmed.ncbi.nlm.nih.gov/23803706/>; Reich et al, 2013, <https://pubmed.ncbi.nlm.nih.gov/23825116/>.
- I fear that the leave-one-out cross-validation procedure may not be sufficient to prevent overfitting. What about leaving out more (consecutive) weeks at a time, e.g. an entire season? Another alternative would be to leave one country/setting at a time.
- What is the effect magnitude of confounders in the multivariate regression analysis?

MINOR COMMENTS

L77: this definition of viral interactions seems quite narrow: could interactions affect onward transmission or disease?

L117: please remove "interactive".

Fig. 3 caption: Please revise the description, there are multiple errors here.

L322: It should be Fig S10 not S11

Version 1:

Reviewer comments:

Reviewer #2

(Remarks to the Author)

The authors have satisfactorily responded to my prior questions. I have no further comments.

Reviewer #3

(Remarks to the Author)

I am happy with the revised manuscript.

Reviewers' comments:

Reviewer #1 (Remarks to the Author):

1: Overall

In this study, the authors aimed to quantify the interactions of SARS-CoV-2, influenza, and RSV at the population level and their impact on transmission dynamics. They used advanced statistical models to assess the exposure-lag-response associations between these respiratory pathogens. Additionally, they simulated the impact of viral interactions and R_e on co-epidemics. The authors found that Influenza A reduces the risk of SARS-CoV-2 infection, while Influenza A and RSV interact to enhance their epidemic peaks.

2: Comments

Introduction

- Overall, the introduction is somewhat lengthy. The last paragraph can be shortened to a few sentences since the details will be explained in the methods section.

Response to reviewer: Thanks for the helpful suggestion. In the revised manuscript, we have shortened the last paragraph of the Introduction to a concise summary of the study objectives and contributions, while moving the methodological details to the Methods section as appropriate.

- The third paragraph lacks a clear focus. It describes different studies but does not clearly outline the study design, the population under study, the location, or the time period within the pandemic.

Response to reviewer: Thank you for your helpful comment. In the revised manuscript, we have clarified the study design, population, location, and time period for each piece of evidence cited in the third paragraph.

- Is this the first population-based study aiming to assess interactions between different viruses and the impact of co-epidemic dynamics? This approach seems quite different from the studies referenced in the introduction, so a more explicit introduction to this concept is needed. Specifically, how interactions could shape the course of future respiratory outbreaks appears somewhat abrupt.

Response to reviewer: Thank you for this insightful comment. We would like to clarify that our study is not the first population-based investigation exploring viral interactions and their impact on co-epidemic dynamics. For example, a 2019 study published in PNAS (Nickbakhsh et al., 2019) used a mathematical modeling approach based on population-level surveillance data to assess how interactions between influenza viruses and common cold may influence patterns of co-circulation. However, to the best of our knowledge, none of the previous research has applied a similar population-based

modeling framework to investigate the interactions between SARS-CoV-2 and other respiratory viruses such as influenza or RSV.

We also appreciate the suggestion regarding the introduction. In our revised manuscript, we have refined the first paragraph to more clearly define viral interactions and elaborate on how such interactions—although occurring at the individual level—can lead to substantial and collectively effects on population-level transmission dynamics. We have also added a clearer transition to the importance of studying these dynamics in the context of future respiratory outbreaks (lines 108-128).

•Line 73: PIT theory—please write this abbreviation in full.

Response to Reviewer: Thank you for your helpful comment. In the revised manuscript, we have removed the use of the abbreviation “PIT” and rephrased the relevant sentence for improved clarity and logical coherence. While the concept from the pathogen invasion theory (PIT) is still referenced, we now cite the corresponding literature directly without explicitly referring to the theory by name (lines 112-116).

•Lines 86-92: This sentence is very long and difficult to read. Please split it into 2-3 sentences.

Response to Reviewer: Thank you for the suggestion. We have revised the sentence by splitting it into two shorter sentences to improve clarity (lines 132-138).

•Line 88: Was the data from Hong Kong and Canada, or was the study conducted there? This needs clarification.

Response to Reviewer: Thank you for the comment. We have clarified in the revised text that the mathematical modeling study used data from Hong Kong and Canada (line 134).

•Line 89: A household study is always in humans. What was the study population—specific households, or only those with children?

Response to Reviewer: Thank you for the comment. The household study was conducted across all age groups. Households with two or more members were included if 80% or more of the members consented to participate. We have clarified this in the revised text (lines 136-138).

•Line 94: Is SARS-CoV-2 still considered "novel"? Consider revising.

Response to Reviewer: Thank you for the suggestion. We have removed the term “novel” and revised the sentence to refer to “SARS-CoV-2” directly (line 141).

•Lines 130 and 145: What does "additional file" refer to? Does this mean supplementary information?

Response to Reviewer: Yes, "additional file" refers to the supplementary information provided with the manuscript. We have clarified this in the revised version (lines 183 and 199).

•Supplementary information, Line 27, and Manuscript, Line 126: The main text states that data is from seven different countries/regions, while the supplementary information mentions six. Please clarify or correct this inconsistency.

Response to Reviewer: Thank you for pointing this out. This was a typo in the supplementary information. It should be seven countries/regions, and we have corrected it accordingly (Supplementary information, Line 29).

Methods

•Please provide references for the data sources used, if available.

Response to Reviewer: Thank you for the comment. The references for the data sources used in each region are listed in the supplementary information.

•Lines 149-151: Are the SARS-CoV-2 variants defined by date, or was strain variant information available in the dataset?

Response to Reviewer: The SARS-CoV-2 variant data were collected on a weekly basis and obtained from the publicly available GISAID dataset (<https://gisaid.org/hcov19-variants/>).

•Is there information available on SARS-CoV-2 and influenza vaccination coverage/uptake? Vaccination status might impact interactions between the viruses and epidemic dynamics.

Response to Reviewer: Thank you for the valuable comment. We agree that vaccination coverage may influence viral interactions and epidemic dynamics. However, weekly vaccination coverage data for SARS-CoV-2, IAV and RSV were not available across all included regions during the study period. Therefore, we used the median positivity rate from the previous year as a proxy of the impact of population immunity on the transmission in the current year, as noted in Lines 262-263 of the main text.

Results

Seasonality

•What are the positivity rates found for the different viruses? Please provide some numerical values (ranges) in the text, as it is difficult to extract this from the figures.

Response to Reviewer: Thank you for the helpful suggestion. We have now added the numerical ranges of weekly positivity rates for each virus in the main text to enhance clarity. Specifically, we reported that IAV ranged from 0% to 67.5%, RSV from 0% to 32.6%, IBV from 0% to 14.7%, and SARS-CoV-2 from 0.4% to 57.2%, based on data across all study regions and periods (see Lines 426-434).

Interactions between Viruses

- The methods state that analyses were restricted to influenza and RSV epidemic seasons. However, in Lines 275-284, results are presented including all data. I suggest removing this part.

Response to Reviewer: While we agree that epidemic-season analyses are crucial for minimizing background noise and highlighting virus–virus interactions during peak transmission periods, we believe that the full-year analysis also provides valuable insights and should be retained. To address both perspectives, we presented the full-year analysis results in Lines 445-458 and the epidemic-season analysis in Lines 500-509. Meanwhile, we have revised the description in the Methods section (lines 313-317) to clarify this. Notably, only the association between IAV positivity and subsequent SARS-CoV-2 infection risk showed a significant negative relationship.

To further investigate this finding and enhance the robustness of our conclusions, we added a new analysis in the revised manuscript by applying a two-pathogen, meta-population mechanistic transmission model to the IAV/SARS-CoV-2 pair using observed infection data in Beijing (lines 549-579). This modeling approach allowed us to more rigorously quantify their interaction dynamics and substantiate the statistical associations observed.

- Line 279: Please refer to supplementary materials in a consistent manner.

Response to Reviewer: Thank you for pointing this out. We have revised the manuscript to ensure that all references to the supplementary materials are now consistent throughout the text.

- There appears to be no reference to Fig. S2 in the manuscript.

Response to Reviewer: Thank you for pointing this out. We have added a reference to Fig. S2 in the revised manuscript at Line 447-458.

- There appears to be no reference to Fig. 3A, 3D, and 3G in the manuscript.

Response to Reviewer: Thank you for your comment. In the revised manuscript, we have updated Fig. 3 by removing the IAV/RSV virus pair, which had wide confidence intervals

and is now presented separately in Fig. S10. References to the updated Fig. 3 have been included in Lines 503-509.

Co-transmission Pattern Shaped by Viral Interaction and Re

•Line 345: "Which also has been observed in some countries"—this statement should be included in the discussion rather than the results.

Response to Reviewer: Thank you for your comment. In the revised manuscript, we replaced the original simulation analysis with a mathematical model fitted to observed infection data (Lines 549-579). As a result, the previous sentence ("which also has been observed in some countries") has been removed.

Discussion

•Line 373: "Our DLNMs further suggest that the strongest suppressive effect on SARS-CoV-2 occurred 4-6 weeks after an IAV infection." I do not fully understand how to interpret this. Do the results not reflect a decreased risk of SARS-CoV-2 infection as influenza incidence increases, with a delay of 4-6 weeks? As it is currently written, it appears that the data is interpreted as if there is a secondary infection (co-infection) within an individual, whereas the data is at the population level. Please clarify.

Response to Reviewer: Thank you for this insightful comment. We agree that the original wording might cause confusion. As our analyses were conducted at the population level using time-series data, the results reflect an association between increased population-level IAV activity and a reduced risk of SARS-CoV-2 infections among populations with a delay of 4–6 weeks, rather than individual-level co-infections.

In the revised manuscript, we have rephrased the sentence for clarity (Lines 630-632), now reading: "*Our DLNM results suggest an association between increased population-level IAV activity and a reduced risk of SARS-CoV-2 infections among populations, with a delay of approximately 4–6 weeks*".

•The specific time span of the lower risk of SARS-CoV-2 after influenza infection requires further discussion. The manuscript references experiments in mice, but a brief look at the paper it seems that the mice only have been observed for 14 days. How do you explain that this effect is visible 4-6 weeks after an initial influenza infection and not immediately? I would expect to see this effect in the early period after primary infection due to an initial immune response boost. This comment is also related to the previous one.

Response to Reviewer: Thank you for your thoughtful comment. The referenced mouse study was conducted at the individual level in animals, while our analysis is based on population-level time-series data. Therefore, the observed delay in SARS-CoV-2 risk reduction likely reflects broader ecological and transmission dynamics, rather than immediate post-infection immune responses. Park et al [Ref 1] found infection with one pathogen at the individual level may reduce the susceptible population at the community

level due to niche competition, thereby altering the overall transmission pattern of another pathogen. This ecological interaction may manifest not as an immediate effect but as a shift in the timing or intensity of subsequent pathogen circulation at the population scale. A similar phenomenon was observed in France during the 2009 H1N1 pandemic, where widespread rhinovirus circulation in the autumn significantly delayed the onset of H1N1 influenza by approximately four weeks [Ref 2]. This supports the idea that interactions between viral respiratory pathogens can influence epidemic timing on a population scale, beyond the window of acute immune responses. We have further clarified this point in the manuscript (Lines xx) to better articulate this distinction between individual-level immune effects and population-level transmission dynamics.

[Ref 1] Park, S. W., Cobey, S., Metcalf, C. J. E., Levine, J. M. & Grenfell, B. T. Predicting pathogen mutual invasibility and co-circulation. *Science* 386, 175–179 (2024).

[Ref 2] Casalegno, J. S. et al. Rhinoviruses delayed the circulation of the pandemic influenza A (H1N1) 2009 virus in France. *Clin. Microbiol. Infect.* 16, 326–329 (2010).

•Line 376: The reference from 1959 is quite outdated. Consider citing a more recent study.

Response to Reviewer: Thank you for your comment. We have replaced it with a more recent and relevant study published in 2018 (line 635).

•Lines 402-410: Is there additional literature that supports your findings?

Response to Reviewer: Thank you for your comment. We have removed the simulation-based analysis from the results section and accordingly deleted the related discussion in Lines 686-694. Instead, we now focus on the mathematical model fitting based on real-world infection data. The revised discussion reflecting these changes can be found in Lines 649-663.

•Lines 408-410: How can policymakers account for these interactions? Please elaborate.

Response to Reviewer: Thank you for the insightful comment. We have elaborated on the public health implications of the observed viral interactions in the revised manuscript (Lines 657-663). Specifically, we note that incorporating pathogen–pathogen interactions into epidemic models could enhance forecasting accuracy and inform more effective preparedness strategies. For example, early influenza activity may create a temporary period of reduced susceptibility to SARS-CoV-2, which could help policymakers predict the potential subsequent waves of other respiratory infections, optimize the timing of vaccination campaigns, implement targeted non-pharmaceutical interventions (NPIs), and allocate healthcare resources more efficiently.

•Line 413: Variability in what? This needs clarification.

Response to Reviewer: Thank you for the comment. The term 'variability' refers to differences in the populations sampled for virological testing across regions. For example, in Beijing, positivity rate data are based on influenza-like illness (ILI) cases, whereas in other regions, the tested populations include all patients seeking care for respiratory symptoms, regardless of ILI criteria. These differences in case definitions and sampling strategies could lead to inconsistencies in the measured positivity rates. However, to address this issue, we standardized the positivity rates within each region to reflect relative levels of circulation of SARS-CoV-2, influenza, and RSV. This approach mitigates the impact of absolute differences in testing coverage or baseline activity levels across regions. Additionally, our analysis did not involve direct comparisons of positivity rates between regions, further reducing potential bias from such variability. We have clarified this point in the revised manuscript (Lines 697-701).

•Lines 433-434: "For example, during IAV peaks, there may be reduced pressure on SARS-CoV-2 control efforts." This statement seems somewhat outdated, as there are currently no SARS-CoV-2 control measures in place.

Response to Reviewer: Thank you for your comment. We agree that the original statement may no longer reflect the current epidemiological context for SARS-CoV-2. To better reflect the current context and highlight the broader public health relevance, we have revised the sentence as follows: *"For example, when predicting short-term trends of pathogen circulation, the influence of interacting pathogens should be taken into account to improve forecasting accuracy. Similarly, vaccination strategies may benefit from incorporating knowledge of viral interactions to support coordinated prevention of multiple pathogens and optimize the effectiveness of immunization programs."* This revision has been made in Lines 722-726.

Final Remarks:

This is an interesting study that provides valuable insights into interactions between SARS-CoV-2, influenza, and RSV at the population level. However, the manuscript requires some clarifications and structural improvements. Additionally, a more detailed discussion on the timing of the observed effects and their biological plausibility is needed.

Response to Reviewer: We sincerely thank the reviewer for the positive feedback and constructive suggestions. In response, we have revised the manuscript to improve clarity and structure. Specifically, we have provided additional explanation regarding the timing of observed viral interactions and discussed the potential biological mechanisms underlying these effects in the Discussion section. We believe these revisions have strengthened the manuscript and better contextualized our findings.

Reviewer #2 (Remarks to the Author):

Thank you for the opportunity to review this manuscript. This study investigates the population-level interactions among SARS-CoV-2, influenza A and B (IAV and IBV), and respiratory syncytial virus (RSV) using surveillance data from seven locations between October 2021 and May 2024. By employing distributed lag nonlinear models within a Bayesian spatiotemporal framework and a mechanistic transmission model, the authors quantified the effects of viral co-circulation on disease dynamics. Their findings indicate that IAV reduces the subsequent risk of SARS-CoV-2 infection, while RSV and IAV exhibit a bidirectional positive interaction, mutually amplifying their epidemic peaks. However, I am not fully convinced that the conclusions are adequately supported by the results.

Major Comments:

1. Figures 2 and 3: I am not fully convinced that the results demonstrate a strong interaction between SARS-CoV-2, influenza, and RSV. The nonlinear changes across seasons and variations between countries suggest that other factors may be influencing the observed patterns. Given that seasonality varies across geographic locations, further clarification is needed on how these differences impact viral interactions. The study also identifies a bidirectional positive interaction between IAV and RSV, but the underlying mechanisms remain unclear—whether due to immune-mediated effects, shared risk factors, or co-infection dynamics.

Response to reviewer: We appreciate your insightful comments. We agree that other factors could influence the observed patterns. To address this concern, we have taken several steps to strengthen the evidence for interactions between the viruses.

First, in order to detect potential interactions between viruses while accounting for confounding factors, we adopted a two-stage statistical modeling approach. In the first stage, we constructed models separately for each region using local data, incorporating multiple potential confounders such as temperature, humidity, non-pharmaceutical interventions (NPIs), and pre-existing immunity levels. This allowed us to obtain more reliable and region-specific estimates of viral associations. In the second stage, we conducted a meta-analysis to synthesize the interaction estimates from individual regions, resulting in a pooled summary measure that reflects overall interaction patterns across diverse epidemiological contexts (see methods section, lines 251-317).

Then, in this revision, we further validated the findings by developing a two-pathogen, meta-population mechanistic transmission model based on observed infection data from Beijing. This model explicitly incorporates interaction terms between viruses to examine how such interactions influence disease transmission patterns. By reconstructing the epidemic processes through model inversion, we were able to derive interaction-related parameters in a mechanistic framework, thereby strengthening the credibility of our findings (see methods section, lines 319-403).

Regarding the bidirectional positive interaction between IAV and RSV, we appreciate the reviewer's interest in the underlying mechanisms. Based on literature review, we found that Sedeyn et al [Ref 3] reported that the non-structural proteins NS1 and NS2 of RSV can cooperatively suppress host innate immune responses, particularly by inhibiting the production and signaling of type I and type III interferons (IFNs), potentially facilitating coinfections with other viruses such as IAV. We have included this discussion in the revised manuscript to offer a plausible explanation for the observed interaction (lines 671-673).

However, we would also like to note that the meta-analysis results across all study regions showed relatively weak and imprecise associations between IAV and RSV, with wide confidence intervals. Therefore, in the mathematical model, we did not further explore this virus pair in detail, focusing instead on more robustly supported interactions.

[Ref 3] Sedeyn, K., Schepens, B. & Saelens, X. Respiratory syncytial virus nonstructural proteins 1 and 2: Exceptional disrupters of innate immune responses. *PLoS Pathog.* 15, e1007984 (2019).

2. What are the definitions of non-epidemic seasons and epidemic seasons? The exclusion of non-epidemic season data to the statistical analysis raises questions about how this decision impacts the interpretation of viral interactions. It is unclear whether sensitivity analyses were conducted to assess the effect of including non-epidemic season data and how the epidemic season was defined for each virus.

Response to reviewer: Thank you for pointing out the need for clarification on seasonal definitions and their implications for our analysis. In our study, all seven regions are located in the Northern Hemisphere. For IAV and RSV, both viruses exhibit very similar seasonal patterns (figure 1), with clear winter peaks and near-zero activity in the summer. Accordingly, we defined the epidemic season as October to March each year for both IAV and RSV [Ref 4].

For SARS-CoV-2, a distinct seasonal pattern has not yet been established. However, upon examining its circulation trends across our study regions, we found that a consistent level of viral activity was maintained year-round, and even during low positivity rate periods, the positivity rate did not drop close to zero.

To address potential concerns about seasonal variation, we conducted statistical analyses based on both the full study period and a subset that included only epidemic seasons. The rationale for excluding non-epidemic season data is to minimize noise from periods of minimal or undetectable virus activity, which often contain large numbers of zeros or background-level signals. Including such data may obscure true associations and reduce the power to detect meaningful interactions. We have added an explanation in the Methods section (lines 313-317).

We found a stable nonlinear negative association between IAV positivity rates and subsequent SARS-CoV-2 infection risk in both the full-year and epidemic-season statistical models, suggesting a consistent interaction pattern. In the revised manuscript, we also added mathematical model fitting for the real infections of IAV and SARS-CoV-2 pair to further validate this interaction (lines 549-479).

Model fitting revealed that IAV infection substantially reduced infection to SARS-CoV-2 by 94.24% (95% CI: 88.50%–99.24%), with the protective effect lasting approximately 38.24 days (95% CI: 35.50–41.29 days). Conversely, SARS-CoV-2 infection was associated with a slight increase of 7.91% in infection to IAV (95% CI: –1.34% to 16.87%), with a duration of protection of around 3.23days (95% CI: 1.13-6.30days).

Fig. 4 Fitted transmission dynamic model with interactions between IAV and SARS-CoV-2 in Beijing. (A) IAV. (B) SARS-CoV-2. Black lines represent the observed incidence, red lines represent the estimated incidence of IAV, blue lines represent the estimated incidence of SARS-CoV-2. Shaded areas representing 2.5–97.5th quantile.

However, for IAV and RSV, the associations were not statistically significant in either the full-year or epidemic-season models; therefore, we did not proceed with further validation using mathematical modeling.

[Ref 4] Szablewski, C., Daugherty, M. & Azziz-Baumgartner, E. Influenza. <https://wwwnc.cdc.gov/travel/yellowbook/2024/infections-diseases/influenza> (2024).

3. The observed asymmetrical interaction between SARS-CoV-2 and IAV suggests that increasing IAV positivity reduces the risk of SARS-CoV-2 infection, yet no evidence was found linking SARS-CoV-2 positivity to subsequent IAV infection. This raises the question

of whether external factors, such as vaccination campaigns or public health interventions, influenced these dynamics. Additionally, potential confounders, including variations in testing rates and reporting delays, should be considered.

Response to reviewer: Thank you for your thoughtful comment. In our analysis, we have carefully controlled for key potential confounders, including temperature, humidity, NPIs, and pre-existing immunity levels, which are known to influence respiratory virus transmission. These variables were incorporated into our statistical models as covariates, ensuring that the estimated viral interaction effects are not spuriously driven by environmental or policy-related factors (lines 252-285). Therefore, we believe that the quantitative results derived from the statistical models are relatively robust.

Regarding variations in testing rates and potential reporting delays, we addressed these concerns in two ways. First, during the data inclusion process, we excluded countries with low weekly sample sizes or poor testing coverage. Specifically, we retained only countries where weekly detection data available for more than 90% of the study period, with weekly sample sizes over 200 per virus (see Supplementary information lines 27-29). Second, our outcome variable in the statistical models was the weekly positivity rate (i.e., proportion of positive tests), rather than the absolute number of positive cases, which helps to mitigate bias introduced by differences in testing volume or reporting lag.

In addition, to further assess the robustness of our findings, we developed a two-pathogen, meta-population mechanistic transmission model using observed infection data from Beijing. The results obtained from this mechanistic model are consistent with those from the statistical analysis, further supporting the validity of our findings.

4. I would suggest that the authors discuss why the nonlinear relationship is observed. For example, the lowest risk of SARS-CoV-2 occurs approximately one month after IAV positivity reaches the 75th percentile. I was wondering what factors contribute to this nonlinear change—specifically, why the lowest risk is achieved after IAV positivity reaches the 75th percentile. How is this related to experimental evidence?

Response to reviewer: Thank you for your suggestions. Our study is based on population-level time-series data, and the observed nonlinear relationship reflects two key aspects: (1) a time lag in the effect of IAV activity on SARS-CoV-2 risk, and (2) a threshold effect associated with IAV activity levels, particularly around the 75th percentile.

The observed time lag may be due to the biological mechanism by which IAV infection induces interferon responses, which can transiently suppress SARS-CoV-2 infection by reducing host susceptibility [Ref 5]. At the population level, this reduction in susceptibility accumulates over time, gradually slowing SARS-CoV-2 transmission. According to the Pathogen Invasion Theory (PIT) proposed by Park et al. [Ref 6], a sufficient level of prior infections is required for one virus to competitively inhibit another by reducing the

available ecological niche—in this case, the susceptible population. Once IAV activity reaches a certain threshold (e.g., the 75th percentile), the ecological competition becomes strong enough to prevent SARS-CoV-2 from successfully spreading.

In addition, behavioral factors (e.g. mobility and contact) may contribute to this nonlinear pattern. Seasonal influenza epidemics are typically accompanied by increased public awareness and behavioral changes—such as mask-wearing, reduced social interactions and mixture, and improved hygiene practices—driven by media coverage and public health messaging. These behaviors, although prompted by influenza, can also reduce the transmission of SARS-CoV-2, contributing to the observed delayed and nonlinear interaction.

Currently, available experimental studies are mostly based on in vitro models and have primarily focused on how the order of viral infections affects replication dynamics. However, to our knowledge, no studies have yet systematically examined how varying doses of IAV exposure influence subsequent SARS-CoV-2 replication. Further experimental research is needed to validate and clarify these interaction mechanisms.

[Ref 5] Gilbert-Girard, S. et al. Viral interference between severe acute respiratory syndrome coronavirus 2 and influenza A viruses. *PLOS Pathog.* 20, e1012017 (2024).

[Ref 6] Park, S. W., Cobey, S., Metcalf, C. J. E., Levine, J. M. & Grenfell, B. T. Predicting pathogen mutual invasibility and co-circulation. *Science* 386, 175–179 (2024).

5. Note that an out-of-season outbreak of RSV was observed in the US during the 2021/22 season. How does this influence model results?

Response to reviewer: Thank you for this insightful comment. We acknowledge that an out-of-season RSV outbreak occurred in the United States during the summer of 2021, likely driven by the relaxation of COVID-19 mitigation measures and shifts in population-level immunity. However, our study period starts in October 2021, which is after the peak of this atypical summer surge. From that point onward, RSV activity followed a more typical seasonal pattern, with peaks observed in the fall and winter months. Therefore, the out-of-season outbreak in summer 2021 is unlikely to have significantly influenced our model results.

6. Figure 4: The behaviour of the interactive transmission model is dependent on population structure, yet it is unclear how the model incorporates differences across geographic locations. While the study employs a meta-population framework, it does not specify whether factors such as age distribution, population density, contact pattern. Additionally, the model assumes fixed reproduction numbers (R_e) and immunity durations for each virus, but it is not evident whether these parameters are adapted to reflect regional epidemiological differences. Given that viral transmission dynamics can vary significantly based on local factors such as vaccination coverage and prior immunity, further clarification is needed on how the model accounts for these variations. Sensitivity

analyses assessing the impact of differing population structures on the model's predictions would strengthen the validity of its findings. Moreover, it would be helpful to understand whether the model outputs have been validated against real-world epidemiological data from the studied locations.

Response to reviewer: Thank you for your thoughtful and constructive comment. In our initial manuscript, the results presented in Figure 4 were based on simulated data, under the assumption of a hypothetical population of 20 million, to explore how interactions among the three viruses might shape co-epidemic patterns. However, we agree with the reviewer that such simulations do not fully reflect real-world complexities, particularly regarding population structure and region-specific epidemiological features.

In response, we have substantially revised Figure 4 in the current version of the manuscript. Specifically, we constructed a two-pathogen, meta-population mechanistic transmission model using available surveillance data from Beijing, which allowed us to better estimate the interaction parameters between viruses in an empirical and local setting. The model reproduced the epidemic curves reasonably well and yielded virus–virus interaction parameters that are consistent with the results of our statistical model, thus providing additional empirical support for the robustness of our findings.

We have updated the Methods (lines 319-403) and Results (lines 549-579) sections accordingly and added further explanation in the Discussion to highlight this improvement.

Reviewer #3 (Remarks to the Author):

This manuscript investigates the extent and impact of virus-virus interactions among SARS-CoV2 (SC2), Influenza A virus (IAV) and RSV, using data from multiple locations. In the first part, the authors employ a Bayesian statistical model to estimate interactions among pathogens. In the second part, they define a mechanistic model and use it to show the impact of interactions on epidemic dynamics. The topic of pathogen interactions is an interesting one, but is also riddled with challenges. I believe that this manuscript is well written and suitable for this journal; however, I do have a few concerns that I hope the authors will take into account when revising the article. Please find my comments below:

MAJOR COMMENTS

- My main concern is about the connection between the statistical analysis and the simulation study. As it stands, the latter attempts to recapitulate the impact of statistically significant results found in the former. However, while the motivation for the simulation exercise is justified, the link does not feel sufficiently strong in my opinion. For example, the statistical analysis suggests non-linear relationships (lag- and positivity-wise) between different pairs of viruses; however, it remains unclear whether simulations actually reproduce such suppression/enhancing effects and the lagged dependency with positivity rates. Fig. 2 seems to suggest that pathogens peak at different times when they are not interacting (by the way, I could not find the values of peak times): could such hard-coded shift in seasonality be misunderstood for a displacement induced by virus-virus interactions? What would be the outcome of the DLNM model if applied to simulated data? Would it be able to recover the absence of interactions ($\sigma=0$) and the non-linear dependency with lagged positivity? Finally, it is not entirely clear to me whether the dashed line in Fig. 2 means that the dynamics of one virus is being kept "fixed": I would expect it to change with σ , at least when virus-virus interactions act on both viruses at the same time.

Response to reviewer: Thank you very much for your valuable suggestions. We agree that the connection between the statistical analysis and the simulation study might not be sufficiently strong in the original submission, especially in terms of capturing the non-linear and lagged nature of virus-virus interactions. Our overarching goal was to use the simulation study to explore how empirically identified interactions might influence epidemic dynamics under various epidemiological scenarios. However, we acknowledge that the initial presentation may have given the impression of a disjointed connection between the two approaches.

In the revised manuscript, we have addressed this issue by incorporating a two-pathogen, meta-population mechanistic transmission model fitted to surveillance data from Beijing (lines 549-579). Specifically, we used the direction, strength, and duration of virus-virus interactions estimated from the statistical analysis as priors or references

when specifying parameters in the mechanistic model. This creates a clearer and more systematic bridge between the empirical findings and the simulation framework. Moreover, the revised mathematical model fitted the observed epidemic curves well, which further supports the reliability and consistency of our modeling framework.

Our mathematical model fitting revealed that IAV infection substantially reduced infection to SARS-CoV-2 by 94.24% (95% CI: 88.50%–99.24%), with the protective effect lasting approximately 38.24 days (95% CI: 35.50–41.29 days). Conversely, SARS-CoV-2 infection was associated with a slight increase of 7.91% in infection to IAV (95% CI: –1.34% to 16.87%), with a duration of protection of around 3.23 days (95% CI: 1.13-6.30 days).

Fig. 4 Fitted transmission dynamic model with interactions between IAV and SARS-CoV-2 in Beijing. (A) IAV. (B) SARS-CoV-2. Black lines represent the observed incidence, red lines represent the estimated incidence of IAV, blue lines represent the estimated incidence of SARS-CoV-2. Shaded areas representing 2.5–97.5th quantile.

Beijing was selected for model calibration due to the availability of high-quality incidence data for both IAV and SARS-CoV-2 in this region. In contrast, other regions only had positivity rate data, which may not accurately reflect true infection incidence among populations for the modelling. Moreover, using Beijing data as an external validation case further strengthens the evidence supporting the observed viral interactions.

Regarding Figure 2, we would like to clarify that it summarizes the results from the statistical models fitted to each region's data. It shows the estimated relative risk of virus B at different levels of virus A positivity, across various lag times. The dashed line represents the minimum observed positivity level of virus A in the dataset, which serves as the reference point (RR = 1) for interpreting the estimated effects.

In the revised manuscript, we have also updated both Fig. 2 and Fig. 3 for clarity and completeness. Figure 2 now presents the exposure–response associations based on the full time series, while Figure 3 focuses on the associations restricted to the influenza season. This separation improves the interpretability and internal consistency of the results presented.

Fig. 2 Exposure-response association between viruses. Associations between percentile of IAV positivity and subsequent SARS-CoV-2 infection risk. (A) RR of SARS-CoV-2 associated with different percentiles of IAV positivity across regions. (B) Pooled contour plot of the lag–response association between percentile of IAV positivity and SARS-CoV-2 risk, relative to the minimum IAV percentile. (C) Pooled exposure–response association between percentile of IAV positivity and SARS-CoV-2 risk. (D–F) Associations between

percentile of RSV positivity and subsequent IAV infection risk. (G–I) Associations between percentile of IAV positivity and subsequent RSV infection risk.

Fig. 3 Exposure-lag-response association between IAV and SARS-CoV-2 during epidemic seasons. (A) RR of SARS-CoV-2 associated with different percentiles of IAV positivity across regions. (B) Pooled exposure–response association between percentiles of IAV positivity and SARS-CoV-2 risk. (C) Pooled contour plot of the lag–response association between percentiles of IAV positivity and SARS-CoV-2 risk, relative to the minimum IAV percentile. (D) RR of SARS-CoV-2 lag-response association for 70th percentile of IAV positivity.

- Fig. 2 shows significant variations among countries, which makes me less confident about some findings. For example, Fig. 2E–F do not seem to suggest a consistent relationship between IAV and RSV.

Response to reviewer: Thank you for pointing this out. Indeed, we observed notable heterogeneity in virus–virus interactions across countries, as shown in Fig. 2. This variation may stem from differences in population structure, testing policies, public health interventions, prior immunity, and environmental factors, all of which can influence viral circulation and interactions. To account for this heterogeneity, we employed a two-stage approach: In the first stage, statistical models were fitted separately for each region to allow region-specific estimates. In the second stage, we conducted meta-analyses to synthesize the results and obtain pooled estimates with confidence intervals, as shown in Supplementary Fig. 3.

Regarding the specific example of IAV and RSV (Fig. 2E–F), while some regions showed weak or inconsistent associations, the pooled meta-analysis result suggested a modest but non-significant positive association, with wide confidence intervals. This finding led us

to exclude the IAV–RSV pair from further real-world transmission modeling due to limited robustness.

- In general, I believe that detecting/measuring pathogen-pathogen interactions is extremely difficult and not enough attention is being given to such challenges. I strongly recommend the authors to review relevant work in this area: Shrestha et al, 2011, <https://pubmed.ncbi.nlm.nih.gov/21876665/>; Shrestha et al, 2013, <https://pubmed.ncbi.nlm.nih.gov/23803706/>; Reich et al, 2013, <https://pubmed.ncbi.nlm.nih.gov/23825116/>.

Response to reviewer: Thank you very much for your insightful suggestion. In the revised manuscript, we have taken into account and discussed the inherently challenges of detecting pathogen–pathogen interactions, as emphasized in the studies by Shrestha et al. (2011, 2013) and Reich et al. (2013). These works highlight the statistical and methodological complexities involved in accurately identifying interactions, especially under key methodological difficulties such as confounding by seasonality, shared transmission pathways, and the limitations of aggregated surveillance data. Inspired by these studies, we incorporated a dual-pathogen transmission dynamics model to estimate interaction parameters between IAV and SARS-CoV-2 based on incidence data from Beijing (Supplementary Fig. S1).

Figure S1: Schematic diagram of two-pathogen interactive transmission model. $\{1,2\}$ denotes the infection status of individuals with respect to virus 1 and virus 2. whereby S =susceptible, I = infectious refractory phase, P = noninfectious refractory phase, R = immunity phase, and subscripts 1 and 2 denote the corresponding virus-specific parameters.

Our model explicitly accounts for bidirectional interactions, including the strength and duration of the effect. Model fitting revealed that IAV infection substantially reduced

infection to SARS-CoV-2 by 94.24% (95% CI: 88.50%–99.24%), with the protective effect lasting approximately 38.24 days (95% CI: 35.50–41.29 days). Conversely, SARS-CoV-2 infection was associated with a slight increase of 7.91% in infection to IAV (95% CI: –1.34% to 16.87%), with a duration of protection of around 3.23 days (95% CI: 1.13-6.30 days).

Fig. 4 Fitted transmission dynamic model with interactions between IAV and SARS-CoV-2 in Beijing. (A) IAV. (B) SARS-CoV-2. Black lines represent the observed incidence, red lines represent the estimated incidence of IAV, blue lines represent the estimated incidence of SARS-CoV-2. Shaded areas representing 2.5–97.5th quantile.

To assess the impact of viral interaction on the co-circulation patterns of SARS-CoV-2 and IAV, we simulated the epidemic trajectories under a counterfactual scenario in which IAV exerts no negative interaction on SARS-CoV-2 (Supplementary Fig. 13). We found that the COVID-19 epidemic peaks would occur earlier and be substantially higher in magnitude. Specifically, the summer 2023 peak would advance by approximately two weeks, with a 2.35-fold (range: 1.67-3.15) increase in peak magnitude, while the winter 2024 peak would advance by about six weeks, with a 3.52-fold (range: 2.21-5.09) increase in magnitude. Overall, our findings suggest that IAV may suppress SARS-CoV-2 transmission, delaying its epidemic progression and attenuating peak intensity, thereby reducing the burden on healthcare resources. Moreover, this model-based approach complements our statistical analysis and helps mitigate some of the limitations identified in the aforementioned literature.

These results have been added to both the Methods (lines 319-403) and Results (lines 549-579) sections of the revised manuscript to reflect this modeling approach and its findings.

Figure S13: Simulated epidemic trends of IAV and SARS-CoV-2 without interaction in Beijing. (A) IAV. (B) SARS-CoV-2. Black lines represent the observed incidence, red lines represent the estimated incidence of IAV, blue lines represent the estimated incidence of SARS-CoV-2. Shaded areas representing 2.5–97.5th quantile.

- I fear that the leave-one-out cross-validation procedure may not be sufficient to prevent overfitting. What about leaving out more (consecutive) weeks at a time, e.g. an entire season? Another alternative would be to leave one country/setting at a time.

Response to reviewer: Thank you for your thoughtful comment regarding the cross-validation strategy. In the revised manuscript, we have updated the cross-validation procedure to address concerns about overfitting. Specifically, we implemented a leave-four-weeks-out cross-validation approach, which better captures potential temporal dependencies compared to leaving out a single week. The posterior predictive performance remained robust under this more stringent evaluation (Supplementary Figs. 3-9).

In addition, we performed cross-validation by geographic setting. That is, we trained the model on data from each individual country/region and evaluated its posterior predictive performance accordingly. We also conducted an external validation using data from

Beijing, which was not included in the initial pooled model fitting. The model's predictive performance remained satisfactory across both internal and external validations.

- What is the effect magnitude of confounders in the multivariate regression analysis?

Response to Reviewer: We initially collected a range of potential confounding variables, including gross domestic product (GDP) per capita, Universal Health Coverage (UHC) index, socio-demographic index (SDI), population density, and age structure (i.e., the proportion of the population aged 65 and above). However, due to multicollinearity among these variables, we retained only a subset of core predictors with minimal redundancy for the multivariate meta-regression analysis (Supplementary Table S19). The results showed that population density reduced residual heterogeneity to 87.81%, while the proportion of older adults lowered it to 88.84%. SDI contributed less to model improvement. When all three variables were included in the model, the I^2 value was 87.50%.

To further explore and quantify the interactions between pathogens, we additionally developed a mechanistic transmission model in which interaction terms were explicitly incorporated into the model structure. This allowed us to more precisely characterize the direction, magnitude, and duration of the interference effect of influenza on SARS-CoV-2 transmission.

Table S19: Multivariate meta-regression models for meta-predictors of the effect of IAV infection on SARS-CoV-2 risk.

Model	AIC	Q	df	p	I^2
Baseline	266.36	604.52	60.00	<0.001	90.07
Popu density	401.38	393.90	48.00	<0.001	87.81
SDI	214.68	509.34	48.00	<0.001	90.58
Above65	317.53	430.14	48.00	<0.001	88.84
Popu density + SDI + Above65	398.96	192.02	24.00	<0.001	87.50

Popu density: Population density

SDI: Socio-demographic index

Above65: Population ages 65 and above

MINOR COMMENTS

L77: this definition of viral interactions seems quite narrow: could interactions affect onward transmission or disease?

Response to Reviewer: Thank you for your insightful comment. We agree that the original definition was too narrow. In the revised manuscript, we have broadened the definition of viral interactions to include not only effects on susceptibility to other pathogens but also potential impacts on onward transmission and disease severity. The revised sentence now reads: "*Infection with one pathogen may influence susceptibility, transmissibility, or disease severity to subsequent pathogen infection through mechanisms such as immune activation, ecological niche competition, or behavioral changes, a phenomenon known as pathogen-pathogen interaction*" (lines 112-116).

L117: please remove "interactive".

Response to Reviewer: The word "interactive" has been removed (line 169), and following another reviewer's recommendation, we have also shortened the corresponding paragraph (lines 162-173). The relevant details have been moved to the Methods section.

Fig. 3 caption: Please revise the description, there are multiple errors here.

Response to Reviewer: Thank you for pointing this out. We have carefully revised the caption of Fig. 3 to correct the identified errors and improve clarity.

L322: It should be Fig S10 not S11

Response to Reviewer: We have corrected the figure reference, it now correctly refers to Fig S9 (line 475).